# Recombinase-Aided Amplification Assay for Rapid Detection of Hypervirulent *Klebsiella pneumoniae* (hvKp) and Characterization of the hvKp Pathotype

Chao Yan,[a,b] Yao Zhou,[a,c] Shuheng Du,[a,c] Bing Du,[a] Hanqing Zhao,[a] Yanling Feng,[a] Guanhua Xue,[a] Jinghua Cui,[a] Lin Gan,[a] Junxia Feng,[a] Zheng Fan,[a] Tongtong Fu,[a] Ziying Xu,[a] Qun Zhang,[a] Rui Zhang,[a] Xiaohu Cui,[a] Ziyan Tian,[a] Yujie Chen,[a] Ting Zhang,[b,d] Lei Huang,[e] ⓘ Jing Yuan[a,b]

ᵃDepartment of Bacteriology, Capital Institute of Pediatrics, Beijing, China
ᵇChildren's Hospital Capital Institute of Pediatrics, Chinese Academy of Medical Sciences & Peking Union Medical College, Beijing, China
ᶜKey Laboratory of Resources Biology and Biotechnology in Western China, Ministry of Education, College of Life Sciences, Northwest University, Xi'an, Shaanxi, China
ᵈBeijing Municipal Key Laboratory of Child Development and Nutriomics, Capital Institute of Pediatrics, Beijing, China
ᵉCenter for Infectious Diseases, the Fifth Medical Center of Chinese PLA General Hospital, Beijing, China

Chao Yan and Yao Zhou contributed equally to this work. Author order was determined on the basis of seniority.

**ABSTRACT** Hypervirulent *Klebsiella pneumoniae* (hvKp) is a major human pathogen associated with liver abscess, pneumonia, meningitis, and endophthalmitis. It is challenging to differentiate hvKp from classical *Klebsiella pneumoniae* (cKp) using conventional methods, necessitating the development of a rapid, sensitive, and convenient assay for hvKp detection. In this study, we constructed a recombinase-aided amplification (RAA) method targeting hvKp genes *peg344* and *rmpA*, and also analyzed the pathogenic characteristics of hvKp. We optimized the reaction temperature and system, and evaluated its sensitivity, specificity, and clinical application. The primer and probe sets *peg344*-set1 and *rmpA*-set2 delivered significant fluorescent signals at 39°C with the shortest gene amplification times (sensitivity: 20 copies/reaction). This RAA assay showed no cross-reactivity with 15 other common pathogenic bacteria. Its applicability was confirmed by the evaluation of 208 clinical specimens, of which 45 were confirmed to be hvKp. The sensitivity and specificity of the RAA assay were both 100% compared with real-time PCR as the reference standard. To verify the assay, we also assessed the diversity of molecular characteristics among the hvKp isolates and identified serotype K1 and sequence type ST23 as the dominant clone. Virulence factors *iroN* and *iutA* were highly associated with virulence level. In conclusion, our novel RAA assay is a powerful tool for early diagnosis and epidemiological surveillance of hvKp.

**IMPORTANCE** *Klebsiella pneumoniae* is the most common opportunistic bacterial species and a major threat to public health. Since the 1990s, hvKp has received increasing attention from public health officials and infectious disease specialists. Hypervirulent strains differ from classical strains in terms of phenotypic features and clinical outcomes. It is hard to identify hvKp from cKp using the conventional methods including colony morphology analysis, serum killing assays, mouse lethality assays, string tests, and real-time PCR. In this study, we established a rapid, sensitive and convenient recombinase-aided amplification assay for hvKp detection targeting virulence genes *peg344* and *rmpA*. Our RAA assay provides an important tool for the rapid diagnosis of infectious diseases caused by hvKp, particularly in primary laboratories.

**KEYWORDS** hypervirulent *Klebsiella pneumoniae*, recombinase-aided amplification, rapid detection, *peg344*, *rmpA*

Address correspondence to Jing Yuan, yuanjing6216@163.com, or Lei Huang, huangleiwa@sina.com.

The authors declare no conflict of interest.

*K*lebsiella pneumoniae (*K. pneumoniae*) is a Gram-negative pathogen that causes pulmonary and urinary tract infections, as well as liver abscesses. In 1986, hypervirulent *Klebsiella pneumoniae* (hvKp) strains, which were clearly distinguishable from classical *Klebsiella pneumoniae* (cKp) strains, were first discovered in Taiwan, and then widely reported in other Asian countries (1). Nowadays, they are increasingly recognized in Europe and the USA (2–5). hvKp strains were originally identified as community-acquired isolates of *K. pneumoniae* with capsule serotypes K1 and K2. They are highly invasive and often cause pyogenic liver abscess (6). To date, four major classes of virulence factors have been well characterized in *K. pneumoniae*: the production of hypercapsule, lipopolysaccharide, siderophores, and fimbriae (7).

Traditional methods for hvKp detection include colony morphology analysis, serum killing assays, mouse lethality assays, string tests, and real-time PCR (8). However, these methods are slow to yield results (several hours, 1 to 2 days, or longer). hvKp strains often exhibit virulence plasmids, including pK2044, pLVPK, and pVir-CR-hvKp4, which encode the most well-characterized hvKp virulence factors (9–11). Importantly, virulence genes on plasmids can be used as specific biomarkers for hvKp detection (12–15). Four such virulence genes are regarded as molecular markers for the detection of hvKp: *rmpA* (regulator of mucoid phenotype A), *rmpA*2, *iron* (siderophore for iron acquisition), and *peg344*.

Of the current biomarkers, *peg344*, which encodes a metabolic transporter of unknown function that is located on the inner membrane, has the highest accuracy, sensitivity, and specificity for the detection of hvKp. *rmpA* is also considered a hvKp-specific gene because hypercapsule production may greatly contribute to hvKp pathogenicity. Therefore, we selected *peg344* and *rmpA* as signature genes for rapid detection of hvKp (12).

Recently, recombinase-based isothermal amplification assay–recombinase polymerase amplification (RPA; developed by TwistDx [Cambridge, United Kingdom]) has replaced traditional PCR approaches. Recombinase-aided amplification (RAA; developed by Qitian [Wuxi, China]) is a detection method based on RPA that enables the rapid, specific, and cost-effective identification of multiple pathogens (16). The recombinases UVSX and UVSY from bacteriophage T4 are key components of RPA, while the recombinant enzymes used in RAA (SC-recA, BS-recA, and Rad51) are obtained from *Escherichia coli*. All of these enzymes can bind tightly to primer DNA at room temperature (17). The RAA assay can be performed at a constant temperature range of 38 to 42°C, with results in 10 to 20 min. Currently, RPA is widely used in clinical diagnosis (18, 19). In this study, we aimed to establish a novel RAA assay targeting *peg344* and *rmpA* for the rapid detection of hvKp in clinical specimens.

## RESULTS

**RAA assay optimization: primer, probe, and reaction temperature.** The hvKp-specific genes *peg344* and *rmpA* were chosen as target regions for the design of primers and probes in this study. To optimize RAA assay conditions, we used recombinant plasmids containing *peg344* and *rmpA* as positive templates, and deionized distilled water as a negative template. We compared three sets of primers and probes designed to target the conserved regions of each gene: *peg344*-set1, *peg344*-set2 and *peg344*-set3, and *rmpA*-set1, *rmpA*-set2, and *rmpA*-set3. The fluorescence signal was monitored for 20 min. Using a single temperature (39°C) and reaction system, the most efficient sets of primers and probes for hvKp detection were identified on the basis of reaction speed and amplification efficiency. At the same template concentration ($1 \times 10^3$ copies/$\mu$L) previously used to compare other primer and probe sets, the primer and probe sets *peg344*-set1 and *rmpA*-set2 showed the most rapid amplification of the two target genes (Fig. 1, 2, and Table 1). Furthermore, the fluorescence signals shown in Fig. 2 demonstrated that results could be detected within 10 min. Therefore, *peg344*-set1 and *rmpA*-set2 were selected as the optimal primer and probe sets for hvKp detection in this study.

To identify the optimal reaction temperature for RAA amplification, we varied the range of the reaction temperature within the optimal range for the RAA fluorescence

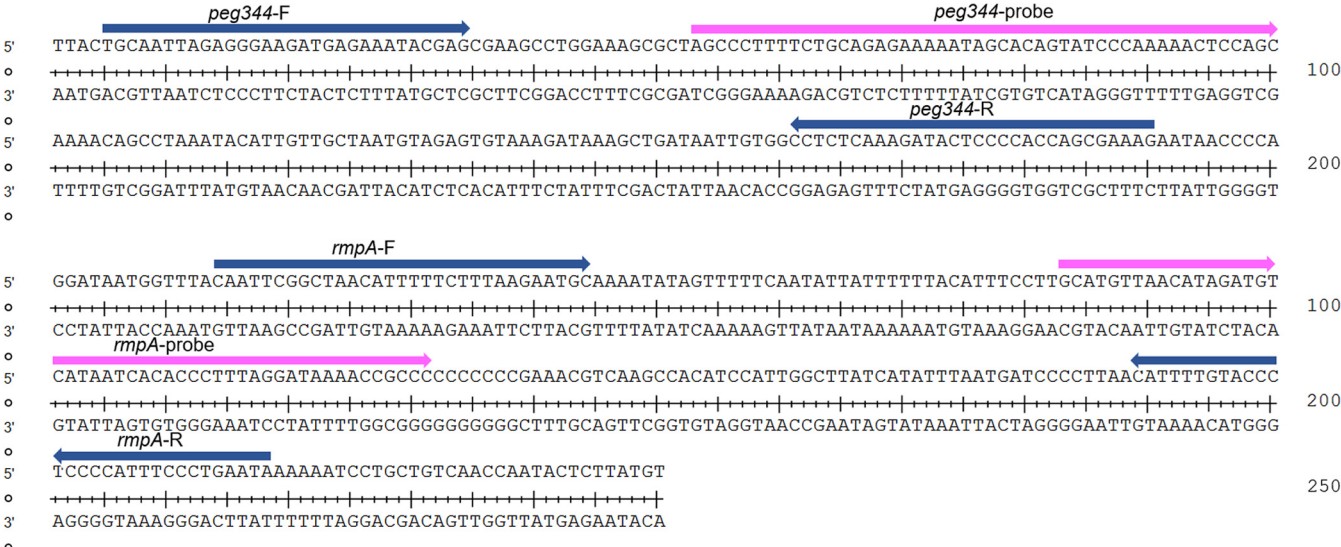

**FIG 1** Selection of specific regions and primer positions for RAA assay. (A) Primer and probe set to amplify the *peg344* gene. (B) Primer and probe set to amplify the *rmpA* gene.

kit (38 to 41°C). The simultaneous fluorescence of both target genes was significantly greater at 39°C than at the other reaction temperatures (38°C, 40°C, and 41°C), and was lowest at 41°C (Fig. 2). Thus, 39°C was selected as the optimal temperature for simultaneous amplification of the two target genes in the RAA assay in this study.

**Analysis of RAA assay sensitivity.** Sensitivity analysis of the RAA assay for hvKp detection was performed using a serial dilution panel of the *peg344* and *rmpA* recombinant plasmids. We used 10-fold dilutions (from $1 \times 10^6$ copies/µL to $1 \times 10^0$ copies/µL) of each plasmid for RAA assays, then compared the results to the output of conventional PCR and real-time PCR. As shown in Fig. 3, replicate dilutions of the recombinant

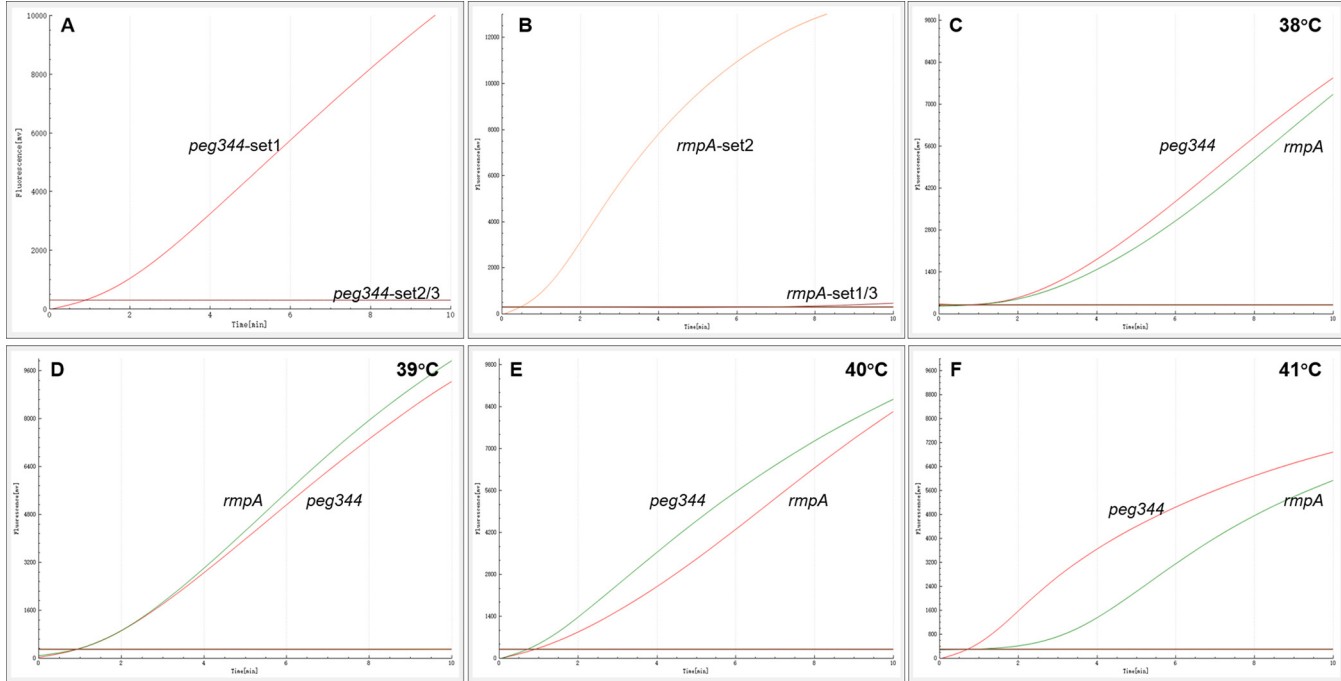

**FIG 2** The most suitable primer and probe sets, and reaction temperature used for RAA analysis. (A) The most suitable primer and probe set for RAA analysis to amplify *peg344* gene. (B) The most suitable primer and probe set for RAA analysis to amplify *rmpA* gene. (C–F) The optimal temperature of RAA reaction. The reaction temperature was 38°C~41°C, respectively.

**TABLE 1** Primers and probes used in this research[a]

| Primer/probe | Sequence (5′–3′) |
| --- | --- |
| Conventional PCR | |
| *peg344*-F1 | GCAATTAGAGGGAAGATGAG |
| *peg344*-R1 | CACGGTAACGCATTAAACGA |
| *rmpA*-F1 | TGTTAGCCGTGGATAATGGT |
| *rmpA*-R1 | CACATCCATTGGCTTATCAT |
| | |
| Real-time PCR | |
| *peg344*-F2 | CCCACCAGCGAAAGAATAAC |
| *peg344*-R2 | AAGGACAGAAAGCCAGTGGA |
| *peg344*-P | FAM-CACGGTAACGCATTAAACGA-BHQ1 |
| *rmpA*-F2 | TTCAGTAGGCATTGCAGCAC |
| *rmpA*-R2 | GGGGCGGTTTTATCCTAAAG |
| *rmpA*-P | FAM-TTTGTTAGCCGTGGATAATGG-BHQ1 |
| | |
| RAA assay | |
| *peg344*-F | TGCAATTAGAGGGAAGATGAGAAATACGAG |
| *peg344*-R | CTTTCGCTGGTGGGGAGTATCTTTGAGAGG |
| *peg344*-probe | AGCCCTTTTCTGCAGAGAAAAATAGCACAG[FAM-dt][THF][BHQ1-dt] CCCAAAAACTCCAGC[3′-block] |
| *rmpA*-F | CAATTCGGCTAACATTTTTCTTTAAGAATGC |
| *rmpA*-R | TATTCAGGGAAATGGGGAGGGTACAAAATG |
| *rmpA*-probe | GCATGTTAACATAGATGTCATAATCACACCC[FAM-dt][THF][BHQ1-dt]AGGATAAAACCGCCC[3′-block] |

[a]FAM, 6-carboxyfluorescein; THF, Tetrahydrofuran; BHQ, Black hole quencher; 3′-block, 3′-phosphate blocker. F, forward primer; R, reverse primer; P, probe.

plasmids (from $1 \times 10^6$ to $1 \times 10^1$ copies/$\mu$L) showed positive fluorescence signals in RAA assays. Therefore, the detection sensitivity of each RAA assay was regarded as 20 copies/reaction. In conventional PCR with primers targeting *peg344* and *rmpA*, the lowest detectable concentration was $1 \times 10^3$ copies/$\mu$L, while that in real-time PCR was 100 copies/$\mu$L.

**Analysis of RAA assay specificity.** We tested the specificity of the RAA assay targeting *peg344* and *rmpA* using each recombinant plasmid and hvKp strain LA.045 as positive controls, and cKp strain ATCC BAA-2146 and deionized distilled water as negative controls. The primer and probe sets for both *peg344* and *rmpA* enabled amplification of recombinant plasmids and hvKp strain LA.045, but did not recognize sequences in the other 15 strains or negative controls (Fig. 4). Therefore, the novel RAA assay demonstrated high specificity for hvKP, with no detectable cross-reactivity with other common pathogens and cKp strain.

**Evaluation of RAA assay using clinical samples and comparison with real-time PCR.** In total, 208 clinical samples, including 158 *K. pneumoniae*-positive samples, collected from healthy individuals ($n = 60$) and inpatients with pneumonia ($n = 80$), bloodstream infections, or liver abscess ($n = 68$) were evaluated. Confirmation of the *K. pneumoniae* isolates was previously performed by 16S rRNA sequencing. RAA results were then compared with the findings from real-time PCR as the reference standard (Table 2). Real-time PCR identified hvKp-positivity in 45 (42 from bloodstream infections or liver abscess patients, 3 from pneumonia patients) of the 208 samples. The RAA assay identified 100% of the 45 hvKp-positive samples, with a kappa value of 1.0 ($P < 0.001$). Compared with real-time PCR, the RAA assay delivered sensitivity of 100% and specificity of 100%.

**Analysis of hvKp phenotype.** To investigate whether the hypermucoviscosity phenotype can be used to define hvKp status, the hvKp isolates were examined by the string test, in which a string length of >5 mm is generally regarded as a positive result. Thirty (66.7%) of the 45 hvKp isolates identified using the RAA assay exhibited large, moist colonies with various degrees of hypermucoviscosity and string lengths >5 mm. Three hvKp isolates surprisingly exhibited colonies with strings >10 cm. The other 15 hvKp isolates were negative for the string test (<5 mm).

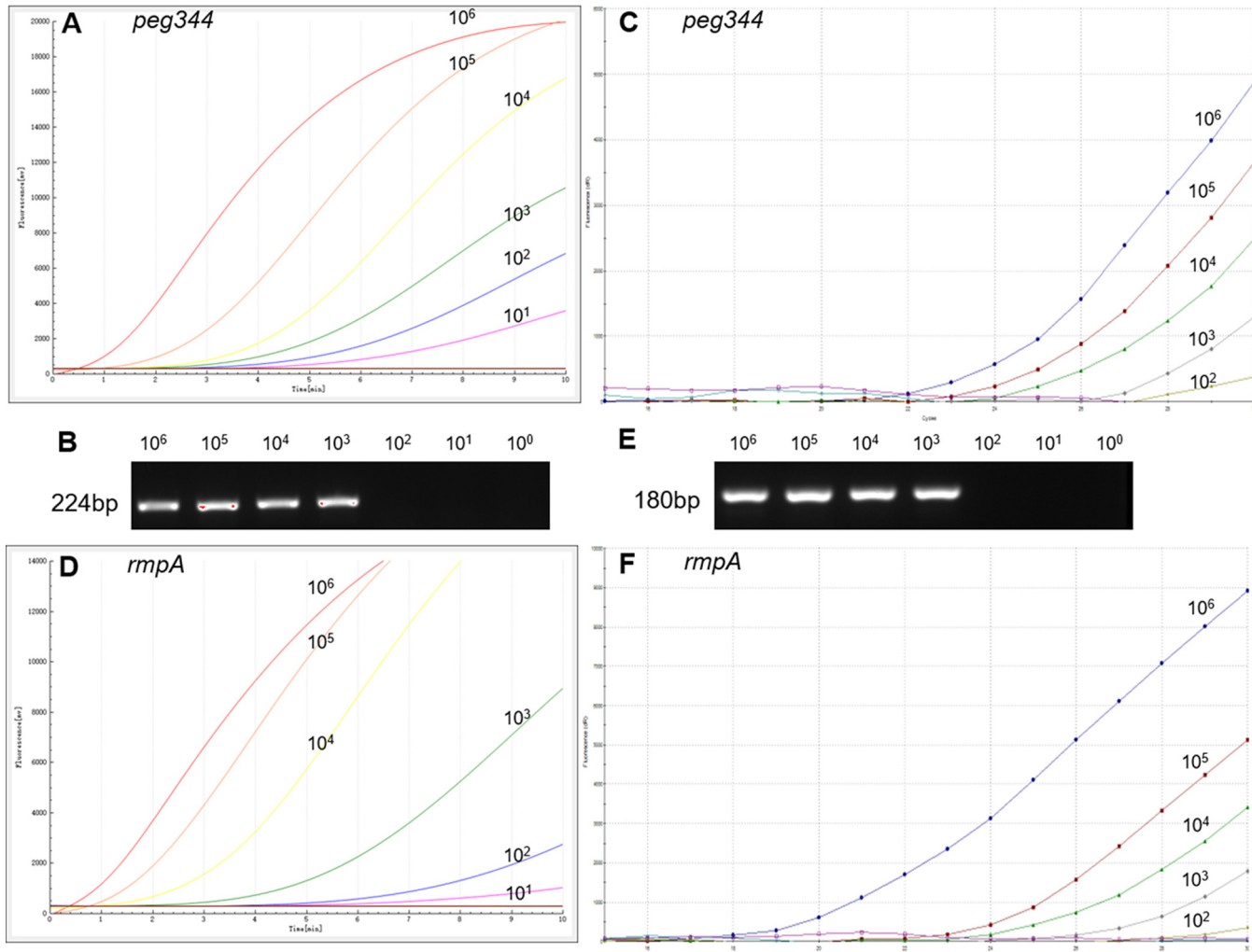

**FIG 3** The sensitivity of RAA assay for hvKp detection. (A and D) The sensitivity of RAA analysis using the primer and probe set *peg344-set1* and *rmpA-set2*. (B and E) The sensitivity of conventional PCR analysis. (C and F) The sensitivity of real-time PCR analysis. The plasmid ranged from $1\times10^{6}$copies/$\mu$L to $1 \times 10^{0}$ copies/$\mu$L.

**Molecular characteristics of hvKp isolates.** Table 3 shows the molecular characteristics (e.g., serotype, MLST type and virulence factors) of the 45 hvKp isolates. Using PCR and sequencing, we identified five distinct serotypes (K1, K2, K20, K54, and K57) among 42 of the isolates. The dominant capsular serotype was serotype K1 (66.7%, 30/45), followed by serotype K2 (15.6%, 7/45), K54 (4.44%, 2/45), K57 (4.44%, 2/45), K20 (2.22%, 1/45), and nontype (6.67%, 3/45). All 30 of the isolates belonging to serotype K1 were also positive for the string test, while the other 15 were negative.

MLST revealed 10 sequence types among the 45 hvKp isolates. The predominant type was ST23 (62.2%, 28/45), followed by ST65 (6.67%, 3/45), ST412 (6.67%, 3/45), ST29 (4.44%, 2/45), ST218 (4.44%, 2/45), ST375 (4.44%, 2/45), ST86 (4.44%, 2/45), ST25 (2.22%, 1/45), ST367 (2.22%, 1/45), and ST420 (2.22%, 1/45). Most (93.3%, 28/30) of the ST23 strains were also serotype K1, while K2 serotype isolates comprised sequence types ST65, ST86, and ST375. Additionally, the 30 string-test positive isolates were typed as ST23, ST25, or ST367.

Except *peg344* and *rmpA*, seven other virulence genes were examined via multiplex-PCR. All virulence genes were present in >60.0% of the hvKp isolates, while *iroN* and *iutA* approached 100 and 97.8%, respectively. Furthermore, 86.7% (26/30) of the hypermucoviscous *K. pneumoniae* isolates harbored all seven virulence genes. Moreover,

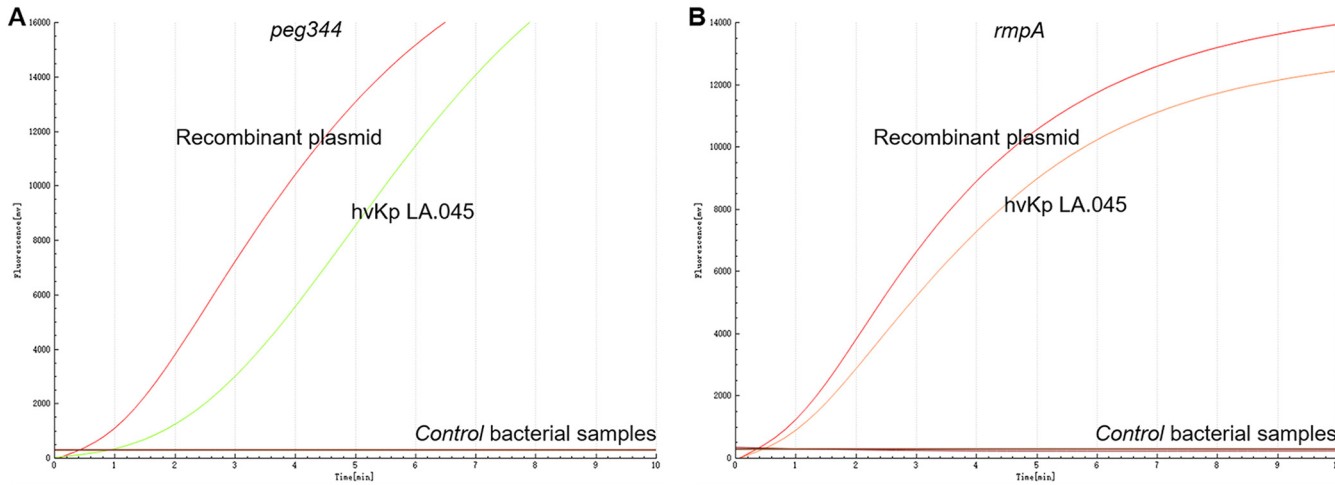

**FIG 4** The specificity of RAA assay for hvKp detection. Only the recombinant plasmids and hvKp strain LA.045 produced amplification signals, whereas the negative control and control bacterial samples (including cKp strain BAA-2146) produced negative amplification signals.

the prevalence of numerous virulence genes in a single strain was notably higher among the serotype K1 isolates than the non-K1 isolates.

## DISCUSSION

*K. pneumoniae* is the most common opportunistic bacterial species and a major threat to public health (1). Since the 1990s, hvKp has received increasing attention from public health officials and infectious disease specialists. Hypervirulent strains differ from classical strains in terms of phenotypic features and clinical outcomes. hvKp can cause community-acquired, aggressive, and metastatic infections that lead to immune dysfunction in patients with diabetes mellitus (e.g., liver abscess, endophthalmitis, meningitis, and septic arthritis) (20, 21). The need to accurately distinguish hvKp from cKp strains is urgent.

Traditional methods for detecting hvKp are too time-consuming to be useful in the assessment of virulence. Nucleic acid detection methods involving traditional PCR and real-time PCR are generally slow and rely on expensive equipment and analyses by specialized researchers. Importantly, such instruments are often cost-prohibitive and unavailable in remote areas, where more rapid detection of hvKp could facilitate efficient clinical diagnosis and treatment. Recombinant enzyme-based isothermal analysis is a simple, rapid, specific, and sensitive nucleic acid amplification method that enables efficient detection. As demonstrated in this study, our novel RAA assay can be performed at a constant temperature of 39°C and completed in 10 min, allowing bacterial detection in a fraction of the time required by other available methods. Furthermore, RAA assays do not require a strict laboratory environment, skilled personnel, or expensive equipment. Additionally, because isothermal amplification does not involve temperature variation, it can be performed with a portable device. Compared with conventional PCR and real-time PCR assays, RAA assays significantly improve detection efficiency (22).

For many years, hvKp classification has been based on hyperviscosity phenotype, specific serotype or MLST, and expression of specific virulence factors. Catalán-Nájera et al. suggested that the hvKp classification should be defined by a key virulence geno-

**TABLE 2** Comparison of recombinant enzyme-assisted amplification methods with real-time PCR and conventional PCR for the detection of hypervirulent *K. pneumoniae*

| Result | RAA *peg344* or *rmpA* gene | Conventional PCR *peg344* or *rmpA* gene | Real-time PCR *peg344* or *rmpA* gene |
|---|---|---|---|
| Positive samples | 45 (Time: 5~10min) | 45 (Time: 2h) | 45 (CT < 30) |
| Negative samples | 163 (no signal) | 163 (no amplicon) | 163 (no peak) |

**TABLE 3** Molecular characteristics of 45 hypervirulent *K.pneumoniae* isolates[a]

| Isolate | Serotype | ST type | Peg-344 | rmpA | magA | allS | iutA | Uge | iroN | kfuBC | ybtA | String test |
|---|---|---|---|---|---|---|---|---|---|---|---|---|
| Isolate hvKp1 | K1 | ST23 | + | + | + | + | + | + | + | + | + | + |
| Isolate hvKp2 | K1 | ST23 | + | + | + | + | + | + | + | + | + | + |
| Isolate hvKp3 | K1 | ST23 | + | + | + | + | + | + | + | + | + | + |
| Isolate hvKp4 | K1 | ST23 | + | + | + | + | + | + | + | + | + | + |
| Isolate hvKp5 | K1 | ST23 | + | + | + | + | + | + | + | + | + | + |
| Isolate hvKp6 | K1 | ST23 | + | + | + | + | + | + | + | + | + | + |
| Isolate hvKp7 | K1 | ST23 | + | + | + | + | + | + | + | + | + | + |
| Isolate hvKp8 | K1 | ST23 | + | + | + | + | + | + | + | + | + | + |
| Isolate hvKp9 | K1 | ST23 | + | + | + | + | + | + | + | + | + | + |
| Isolate hvKp10 | K1 | ST23 | + | + | + | + | + | + | + | + | + | + |
| Isolate hvKp11 | K1 | ST23 | + | + | + | + | + | + | + | + | + | + |
| Isolate hvKp12 | K1 | ST23 | + | + | + | + | + | + | + | + | + | + |
| Isolate hvKp13 | K1 | ST23 | + | + | + | + | + | + | + | + | + | + |
| Isolate hvKp14 | K1 | ST23 | + | + | + | + | + | + | + | + | + | + |
| Isolate hvKp15 | K1 | ST23 | + | + | + | + | + | + | + | + | + | + |
| Isolate hvKp16 | K1 | ST23 | + | + | + | + | + | + | + | + | + | + |
| Isolate hvKp17 | K1 | ST23 | + | + | + | + | + | + | + | + | + | + |
| Isolate hvKp18 | K1 | ST23 | + | + | + | + | + | + | + | + | + | + |
| Isolate hvKp19 | K1 | ST23 | + | + | + | + | + | + | + | + | + | + |
| Isolate hvKp20 | K1 | ST23 | + | + | + | + | + | + | + | + | + | + |
| Isolate hvKp21 | K1 | ST23 | + | + | + | + | + | + | + | + | + | + |
| Isolate hvKp22 | K1 | ST23 | + | + | + | + | + | + | + | + | + | + |
| Isolate hvKp23 | K1 | ST23 | + | + | + | + | + | + | + | + | + | + |
| Isolate hvKp24 | K1 | ST23 | + | + | + | + | + | + | + | + | + | + |
| Isolate hvKp25 | K1 | ST23 | + | + | + | + | + | + | + | + | + | + |
| Isolate hvKp26 | K1 | ST23 | + | + | + | + | + | + | + | + | + | + |
| Isolate hvKp27 | K1 | ST23 | + | + | + | + | + | + | + | + | − | + |
| Isolate hvKp28 | K1 | ST23 | + | + | + | + | + | + | + | + | − | + |
| Isolate hvKp29 | K1 | ST367 | + | + | + | − | + | + | + | + | + | + |
| Isolate hvKp30 | K1 | ST25 | + | + | + | + | + | + | + | − | − | + |
| Isolate hvKp31 | K2 | ST65 | + | + | − | + | + | + | + | + | − | − |
| Isolate hvKp32 | K2 | ST65 | + | + | − | + | + | + | + | − | + | − |
| Isolate hvKp33 | K2 | ST65 | + | + | − | + | + | + | + | − | + | − |
| Isolate hvKp34 | K2 | ST86 | + | + | − | + | + | + | + | − | + | − |
| Isolate hvKp35 | K2 | ST86 | + | + | − | + | + | + | + | − | + | − |
| Isolate hvKp36 | K2 | ST375 | + | + | − | − | + | + | + | − | − | − |
| Isolate hvKp37 | K2 | ST375 | + | + | − | − | + | + | + | − | − | − |
| Isolate hvKp38 | K20 | ST420 | + | + | − | + | + | + | + | + | − | − |
| Isolate hvKp39 | K54 | ST29 | + | + | − | − | + | + | + | − | + | − |
| Isolate hvKp40 | K54 | ST29 | + | + | − | − | + | + | + | − | + | − |
| Isolate hvKp41 | K57 | ST218 | + | + | − | + | + | − | + | − | + | − |
| Isolate hvKp42 | K57 | ST218 | + | + | − | − | + | − | + | − | − | − |
| Isolate hvKp43 | NT | ST412 | + | + | − | + | + | − | + | − | − | − |
| Isolate hvKp44 | NT | ST412 | + | + | − | + | − | − | + | − | − | − |
| Isolate hvKp45 | NT | ST412 | + | + | − | + | + | − | + | − | − | − |

[a]*magA*, mucoviscosity-associated gene A; *allS*, allantoin metabolism; *iutA*, encode the aerobactin; *uge*, UDP galacturonate 4-epimerase; *iroN*, iron system capture; *kfuBC*, iron transport and phosphotransferase; *ybtA*, dsiderophore yersiniabactin. NT, not a K1, K2, K5, K20, K54, or K57 serotype.

type, rather than a high-mucosity phenotype (23). However, only about 17% of hvKp strains are hypermucoviscous, less than the 23% of cKp strains that also possess this characteristic. The correlation between hypermucoviscosity and clinical features observed with hvKp infection is variable, ranging from 51% to 98%. Understandably, the use of this phenotype alone to define hvKp isolates has created confusion. Likewise, the utilization of K1/K2 serotype or ST23 to define hvKp is imperfect. More recently, biomarkers on virulence plasmids have been proved highly accurate for differentiating hvKp from cKp strains. Harada et al. reported that virulence genes such as *rmpA*, *rmpA*2, *peg344*, *iroB*, and *iucA*, as well as quantitative siderophore assays for strains that carry ≥30 $\mu$g/mL iron, can be used to distinguish hvKp strains from cKp strains with >95% accuracy (24). The reported accuracy, sensitivity, and specificity of *peg344* for identifying hvKp strains were 0.97, 0.99 and 0.96, respectively, while those for *rmpA* were 0.96, 0.98, and 0.93. All of the

two virulence genes were associated with a hazard ratio of >25 for severe illness or even death (12). The *peg344* gene is located on a plasmid and encodes an endosomal transporter that Bulger et al. regarded as a key hvKp virulence factor (25). The *rmpA* gene is a critical virulence factor that participates in capsule production and hypermucoviscosity. Thus, we chose *peg344* and *rmpA* as the signature genes for rapid and sensitive detection of hvKp in our novel RAA assay. To ensure assay stability, we assessed the two target genes in hvKp isolates with different characteristics and analyzed *peg344* and *rmpA* gene sequences in the complete genomes of various hvKp strains available in GenBank. To design primers and probes for each gene, we selected transition regions without genetic mutations. In this study, we did not observe cross-reactions with 15 other common pathogens and cKp strain.

Our RAA assay successfully detected the signature genes *peg344* and *rmpA* with high sensitivity. The sensitivity was sufficient to detect 20 copies/reaction of each gene, demonstrating greater sensitivity than that of the real-time PCR assay used for comparison (100 copies/$\mu$L). In 2000, Notomi et al. developed a detection method known as loop-mediated isothermal amplification (26). Yan et al. then reported a loop-mediated isothermal amplification assay that targeted *peg344* to detect hvKp (27); however, the minimum detection concentration was 0.475 pg/$\mu$L and the assay required nearly 1 h for results. Our RAA assay requires less time and demonstrates comparatively greater sensitivity. To evaluate the clinical applicability of our RAA assay, we screened 208 clinical samples using the new assay, as well as real-time PCR and conventional PCR. The agreement among RAA, real-time PCR, and conventional PCR was 100%, suggesting that our novel RAA assay is suitable for the diagnosis of hvKp in clinical samples. The hvKp isolates identified in the present study included five previously reported serotypes (K1, K2, K20, K54, and K57) and 10 MLST variants (ST23, ST25, ST29, ST218, ST367, ST375, ST412, ST420, ST65, and ST86).

The string tests revealed that only 66.7% (30/45) of our hvKp isolates exhibited hypermucoviscosity, demonstrating that the hypermucoviscosity phenotype of *K. pneumoniae* does not equate to hypervirulence. The K1 serotype comprised 66.7% of the hvKP isolates, which was consistent with a previous report (8). Zhang et al. found that the MLST variants ST23, ST268, ST375, ST412, ST660, and ST65 were closely associated with the hvKp classification (28, 29). Guo et al. found that ST23 was the most common MLST variant among hvKp strains (30). In our study, 45 hvKp strains belonged to 10 MLST variants, and 62.2% were ST23. In summary, K1-ST23 was the most prevalent clone of the hvKp isolates. Although we found that virulence genes *iroN* and *iutA* were highly associated with virulence level, other virulence factors may not be useful to verify hvKp status.

This study had some limitations. First, concurrent amplification of multiple target genes in the RAA assay system is potentially problematic because increasing amounts of primers and probes in a single tube can cause nonspecific amplification. Second, our assay involved fluorescence analysis. More convenient methods that enable assessment by the unaided eye should be developed for future RAA assays.

In summary, we developed a rapid RAA assay for the detection of hvKp. This method demonstrated high sensitivity and specificity and does not rely on expensive equipment or analyses by specialized researchers. It is suitable for application in diagnostic laboratories with minimal equipment and may reduce the morbidity and mortality of hvKp infection by supporting clinical treatments and disease control in primary hospitals. Our RAA assay provides an important tool for the rapid diagnosis of infectious diseases and epidemiological surveillance, particularly in primary laboratories.

## MATERIALS AND METHODS

**Bacterial strains and clinical samples.** Seventeen clinical pathogenic bacteria were used to evaluate the specificity of the new RAA method for the detection of hvKp. International standard strain stored in our microorganism center included *Haemophilus influenzae* ATCC10211, *Staphylococcus aureus* ATCC29213, *Staphylococcus epidermidis* ATCC12228, *Streptococcus pneumoniae* ATCC49619, *Escherichia coli* ATCC25922, *Mycoplasma pneumoniae* M129 ATCC29342, *Campylobacter jejuni* ATCC33560, *Pseudomonas aeruginosa* ATCC 27853, *Mycobacterium tuberculosis* ATCC25618, and *Legionella pneumophila* ATCC33823. The *K. pneumoniae* strain ATCC BAA-2146 was used as reference strain for cKp. The *K. pneumoniae* strain LA.045 isolated from

a liver abscess patient and proved to be hypervirulent using mouse lethality assay was used as reference strain for hvKp. Clinical isolates collected from our microorganism center included *Burkholderia cepacia*, *Chlamydia trachomatis*, *Klebsiella aerogenes*, *Klebsiella oxytoca*, *Serratia marcescens*. We obtained 208 clinical specimens collected from healthy individuals ($n = 60$) and inpatients with pneumonia ($n = 80$), bloodstream infections or liver abscess ($n = 68$). The patients came from different provinces in China. Each specimen was obtained from a separate individual. Specimens included bronchoalveolar lavage fluid, throat swabs, stool, blood cultures, and liver aspirate.

This study has been evaluated by the research board of the Ethics Committee of the Capital Institute of Pediatrics in Beijing, China. Patients involved in this study were anonymized, no informed consent was acquired because of the retrospective study.

**DNA extraction.** Total DNA was extracted from the isolates using the QIAamp DNA minikit (Qiagen, Hilden, Germany) following the manufacturer's instructions. DNA was eluted in 50 $\mu$L DNase–free water and stored at −80°C until use.

**Preparation of recombinant plasmids.** The virulence genes *peg344* and *rmpA* were screened for target gene suitability using various sources of data from the National Center for Biotechnology Information (https://www.ncbi.nlm.nih.gov) database. The full nucleotide sequences of *peg344* (903 bp) and *rmpA* (633 bp) from *K. pneumoniae* strain NTUH-K2044-CR plasmid (GenBank accession number: MZ475709) were cloned into the PGM-T vector (Tiangen Biochemical Technology Co., Ltd.).

Plasmid concentrations were determined using a Thermo Scientific Nanodrop Spectrophotometer. Recombinant plasmid preparations were diluted from $10^6$ copies/$\mu$L to $10^0$ copies/$\mu$L and stored at −80°C following quantification. We calculated the copy number of the recombinant plasmid using the following formula: DNA copy number (copy number/$\mu$L) = $(6.02 \times 10^{23} \times$ Plasmid concentration [ng/$\mu$L] $\times 10^{-9})/$(DNA length [nucleotide] $\times 660$).

**RAA primer and probe design.** Target sequences in conserved regions of *peg344* and *rmpA* were manually constructed in accordance with the principles of RAA primer and probe design: primer size of 30 to 35 bp, probe size of 46 to 52 bp, and RAA amplification product size of 100 to 200 bp. The primer and probe candidates for the RAA assay are listed in Table 1. The specificities of primers and probes were analyzed using BLAST software (https://blast.ncbi.nlm.nih.gov/). The online software OligoEvaluator (http://www.oligoevaluator.com/LoginServlet) was used to analyze melting temperature ($T_m$, °C), GC%, secondary structure, and primer dimerization. All primers and probes were synthesized by Biotech (Shanghai) Co., Ltd., and purified by high performance liquid chromatography.

**RAA assay.** We used a commercial RAA kit (Qitian Biological Co., Ltd., Jiangsu, China) for the RAA assay. The 50-$\mu$L reaction mixture contained 25 $\mu$L reaction buffer, 15.7 $\mu$L DNase-free water, 2.1 $\mu$L forward primer, 2.1 $\mu$L reverse primer, 0.6 $\mu$L probe, and 2.5 $\mu$L magnesium acetate. The reaction mixture was added to tubes that contained the RAA enzyme mix in lyophilized form. A 2-$\mu$L aliquot of DNA template was then added to each tube in a separate biosafety cabinet. The B6100 Oscillation mixer (QT-RAA-B6100; Jiangsu Qitian Bio-Tech Co., Ltd., China) was used to mix each tube for 4 min. Finally, a fluorescence detector (QT-RAA-1620; Jiangsu Qitian Bio-Tech Co., Ltd., China) was used to measure fluorescence for 20 min at 39°C.

**PCR and sequencing.** Each 25-$\mu$L PCR mixture contained the following components: 12.5 $\mu$L PCR master mix reagent (Tiangen Biotech Co., Ltd., Beijing, China), 9.5 $\mu$L DNase-free water, 0.5 $\mu$L of 10 $\mu$M forward primer 1 (F1) and reverse primer 1 (R1), and 2 $\mu$L DNA template. PCR cycling conditions were 94°C for 10 min; 35 cycles of 94°C for 30 s, 58°C for 30 s, and 72°C for 45 s; and a final extension at 72°C for 10 min. The primer sets *peg344*-F1 and *peg344*-R1, and *rmpA*-F1 and *rmpA*-R1, were predicted to produce amplicons of 224 bp and 180 bp, respectively. PCR products were separated by electrophoresis in 1.5% agarose gels, then stained with GeneGreen. Images were obtained using a Gel Doc EQ imaging system. Amplified products were sequenced by Sangon Biotech (Shanghai) Co., Ltd. Sequences were analyzed with BLAST software.

**Real-time PCR.** Real-time PCR analysis was used as a reference standard for the detection of hvKp. The oligonucleotide sequences of primers and probes are listed in Table 1. Real-time PCR was performed using *Premix Ex Taq* (Probe qPCR, TaKaRa, Dalian, China) in accordance with the manufacturer's instructions, in a total reaction volume of 25 $\mu$L, including 2.5 $\mu$L DNA template. DNA was amplified using the real-time PCR system, with 40 cycles of 95°C for 15 s and 60°C for 1 min. A positive result was recorded for each sample with a cycle threshold value <30.

**Hypermucoviscous phenotypical identification of hvKp.** hvKp phenotypes were analyzed through the identification of hypermucoviscous strains using the string test. Briefly, individual colonies of hvKp strains cultured on blood agar were touched with an inoculating loop, which was then lifted off the agar surface. A strain was considered mucoid or hypermucoviscous when a string length of >5 mm was observed.

**Determination of serotype, multilocus sequence typing (MLST), and virulence factors of hvKp.** Serotyping of hvKp isolates was determined by PCR using primers specific for detection of serotypes K1, K2, K5, K20, K54, and K57, as described previously (31). MLST of hvKp was performed in accordance with the protocol described on the *K. pneumoniae* MLST website (http://www.pasteur.fr/mlst/). Seven housekeeping genes for *K. pneumoniae* were amplified: *gapA*, *infB*, *mdh*, *pgi*, *phoE*, *ropB*, and *tonB*. Alleles and sequence types were analyzed using the MLST database (https://bigsdb.pasteur.fr/klebsiella/). Seven virulence genes (*magA*, *allS*, *iutA*, *uge*, *kfuBC*, *ironN*, and *ybtA*) were amplified, sequenced, and analyzed by electrophoresis in 1.5% agarose gels (32, 33).

## ACKNOWLEDGMENTS

This work was supported by grants from the National Natural Science Foundation of China (82002191, 82130065, 82272352, and 32170201), Beijing Natural Science Foundation

(7222014), FENG foundation (FFBR 202103), the Research Foundation of Capital Institute of Pediatrics (PY-2019-06 and CXYJ-2021-04), Public service development and reform pilot project of the Beijing Medical Research Institute (BMR2019-11). We thank Michelle Kahmeyer-Gabbe, PhD, from Liwen Bianji (Edanz) (www.liwenbianji.cn) for editing the English text of a draft of the manuscript.

J.Y. and C.Y. designed the study. Y.Z., S.D., B.D., J.C., Q.Z., R.Z., and X.C. performed the experiments. G.X., H.Z., Z.T., and Y.F. collected the clinical strains. Y.Z., S.D., Y.C., L.G., J.F., Z.F., T.F., Z.X., and Q.Z. analyzed the results. C.Y. and Y.Z. wrote the manuscript. J.Y., T.Z., and L.H. revised the manuscript. All authors read and approved the final manuscript.

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
