## [Reviewer comments · Microbiology Spectrum]

Microbiology Spectrum

Recombinase-Aided Amplification Assay for Rapid Detection of Hypervirulent *Klebsiella pneumoniae* (hvKp) and Characterization of the hvKp Pathotype

Chao Yan, Yao Zhou, Shuheng Du, Bing Du, Hanqing Zhao, Yanling Feng, Guanhua Xue, Jinghua Cui, Lin Gan, Junxia Feng, Zheng Fan, Tongtong Fu, Ziyang Xu, Qun Zhang, Rui Zhang, Xiaohu Cui, Ziyang Tian, Yujie Chen, Ting Zhang, Lei Huang, and Jing Yuan

Corresponding Author(s): Jing Yuan, Capital Institute of Pediatrics, and Lei Huang, Fifth Medical Center of Chinese PLA General Hospital

Review Timeline:

Submission Date:	October 4, 2022
Editorial Decision:	January 9, 2023
Revision Received:	February 6, 2023
Accepted:	February 17, 2023

Editor: MARK PANDORI

Reviewer(s): Disclosure of reviewer identity is with reference to reviewer comments included in decision letter(s). The following individuals involved in review of your submission have agreed to reveal their identity: Peng Zhu (Reviewer #2)

Transaction Report:

DOI: <https://doi.org/10.1128/spectrum.03984-22>

January 9, 2023

Prof. Jing Yuan
Capital Institute of Pediatrics
Department of Bacteriology
No. 2 yabao road, Chaoyang District
Beijing, Beijing 100020
China

Re: Spectrum03984-22 (Recombinase-Aided Amplification Assay for Rapid Detection of Hypervirulent *Klebsiella pneumoniae* (hvKp))

Dear Prof. Jing Yuan:

Link Not Available

Sincerely,

MARK PANDORI

Journals Department
While we are willing to consider a revised version of this paper at Spectrum, it would be in your best interest to improve the writing. I recommend that you ask a colleague of yours who is a native English speaker to read and provide you some feedback on the writing. You are also welcome to use one of the services here: <https://journals.asm.org/content/language-editing-services>

Reviewer comments:

Reviewer #1 (Comments for the Author):

Manuscript is sound in methodology and analysis. Appropriate controls were implemented.

Line 74 Missing space between "siderophore" and "for"

Line 133 lists RAA sensitivity as 20 copies while line 239 lists 10 copies

Reviewer #2 (Comments for the Author):

The authors developed a new method to rapidly detect hvKp in clinical samples based on recombinase-aided amplification, and thought this approach is a powerful tool to solve the challenge of timely detection of hvKp infections in clinical and surveillance settings. There are comments which the authors should clarify.

Major comments:

1. Considerable English revision is needed.

2. The title is focused on the rapid detection of hvKp. However, in manuscript, the authors also have written much information about phenotype and molecular characteristics of hvKp. These contents are not in accordance with the topic of this manuscript. Please clarify this.

3. There is on hvKp/cKp standard strains were used to evaluate the specificity in the study. It is not enough to use only plasmid containing genes *peg-344* and *rmpA*, at least two standard strains of hvKp and cKp should also be examined.

4. It is not convincing to evaluate the RAA assay in clinical detection of hvKp only using clinical isolates. I suggest the authors to check the ability of the developed assay from clinical samples, such as, blood, saliva and nasopharyngeal swabs.

5. The authors should analyze the economics of this assay and explain in discussion, since the authors thought the cost of RAA assay is lower than real-time PCR or other nucleic acid-based detection methods in discussion.

Minor comments:

1. Line 108: Delete the "*peg-344* and *rmpA*"

2. Line 114: Delete the "using the RAA assay developed"

3. Line 114-116: Refine and rewrite this sentence.

4. Line 117: Change "The optimal reaction temperature range" to "The range of the reaction temperature".

5. Line 121: Change "Fluorescence" to "Fluorescent".

6. Line 123: Delete "evaluated"

7. Line 126: Add "analysis" before the word "of"

8. Line 130: The results from conventional PCR and real-time PCR could not be reviewed in Figure 3. Suggest the authors add these figures in Figure 3.

9. Line 131: Here, the unit of the concentration of recombinant plasmid is "copies/reaction", which is not in line with "copies/ μ l" in Line 135. Please the authors clarify.

10. Line 141-143: Refine and rewrite this sentence.

11. Line 148: Delete the "assay" before the word "results".

12. Line 149: Delete "which".

13. Line 159-161: Refine and rewrite this sentence.

14. Line 198-199: Refine and rewrite this sentence.

15. Line 202-204: Refine and rewrite this sentence.

Staff Comments:

Preparing Revision Guidelines

For complete guidelines on revision requirements, please see the journal Submission and Review Process requirements at

<https://journals.asm.org/journal/Spectrum/submission-review-process>. **Submissions of a paper that does not conform to Microbiology Spectrum guidelines will delay acceptance of your manuscript. "**

Please return the manuscript within 60 days; if you cannot complete the modification within this time period, please contact me. If you do not wish to modify the manuscript and prefer to submit it to another journal, please notify me of your decision immediately so that the manuscript may be formally withdrawn from consideration by Microbiology Spectrum.

The authors developed a new method to rapidly detect hvKp in clinical samples based on recombinase-aided amplification, and thought this approach is a powerful tool to solve the challenge of timely detection of hvKp infections in clinical and surveillance settings. There are comments which the authors should clarify.

Major comments:

1. Considerable English revision is needed. There are too many chinglish expression in manuscript.

2. The title is focused on the rapid detection of hvKp. However, in manuscript, the authors also have written much information about phenotype and molecular characteristics of hvKp. These contents are not in accordance with the topic of this manuscript. Please clarify this.

3. There is on hvKp/cKp standard strains were used to evaluate the specificity in the study. It is not enough to use only plasmid containing genes *peg-344* and *rmpA*, at least two standard strains of hvKp and cKp should also be examined.

4. It is not convincing to evaluate the RAA assay in clinical detection of hvKp only using clinical isolates. I suggest the authors to check the ability of the developed assay from clinical samples, such as, blood, saliva and nasopharyngeal swabs.

5. The authors should analyze the economics of this assay and explain in discussion, since the authors thought the cost of RAA assay is lower than real-time PCR or other nucleic acid-based detection methods in discussion.

Minor comments:

1. Line 108: Delete the “peg-344 and rmpA”

2. Line 114: Delete the “using the RAA assay developed”

3. Line 114-116: Refine and rewrite this sentence.

4. Line 117: Change “The optimal reaction temperature range” to “The range of the reaction temperature”.

5. Line 121: Change “Fluorescence” to “Fluorescent”.

6. Line 123: Delete “evaluated”

7. Line 126: Add “analysis” before the word “of”

8. Line 130: The results from conventional PCR and real-time PCR could not be reviewed in Figure 3. Suggest the authors add these figures in Figure 3.

9. Line 131: Here, the unit of the concentration of recombinant plasmid is “copies/reaction”, which is not in line with “copies/μl” in Line 135. Please the authors clarify.

10. Line 141-143: Refine and rewrite this sentence.

11. Line 148: Delete the “assay” before the word “results”.

12. Line 149: Delete “which”.

13. Line 159-161: Refine and rewrite this sentence.

14. Line 198-199: Refine and rewrite this sentence.

15. Line 202-204: Refine and rewrite this sentence.

Dear editor,

Thank you for your e-mail informing us that our manuscript will benefit from being revised according to the suggestions of the reviewers.

We would like to thank all the reviewers for their very kind and constructive comments on our manuscript. Our responses to their comments are as follows:

Changes in the revised manuscript are red.

Reviewer comments:

Reviewer #1 (Comments for the Author):

Manuscript is sound in methodology and analysis. Appropriate controls were implemented.

Line 74 Missing space between "siderophore" and "for"

Answer: Thanks for your suggestion. As suggested, we add space between "siderophore" and "for" (Line 82 in manuscript R1).

Line 133 lists RAA sensitivity as 20 copies while line 239 lists 10 copies

Answer: Thanks for your suggestion. As suggested, we revised 10 copies to 20 copies/reaction (Line 237 in manuscript R1).

Reviewer #2 (Comments for the Author):

The authors developed a new method to rapidly detect hvKp in clinical samples based on recombinase-aided amplification, and thought this approach is a powerful tool to solve the challenge of timely detection of hvKp infections in clinical and surveillance settings. There are comments which the authors should clarify.

Major comments:

1. Considerable English revision is needed.

Answer: Thanks for your suggestion. As suggested, we revised English of our

manuscript. We thank Michelle Kahmeyer-Gabbe, PhD, from Liwen Bianji (Edanz) (www.liwenbianji.cn) for editing the English text of a draft of this manuscript. Professor Michelle Kahmeyer-Gabbe's research areas include Microbiology, Biochemistry and Cell Biology, Complementary and Alternative Medicine. We hope that this manuscript is now conducive to readers' reading and understanding. We also uploaded the EdanzEditing Certificate as supplementary materials.

2. The title is focused on the rapid detection of hvKp. However, in manuscript, the authors also have written much information about phenotype and molecular characteristics of hvKp. These contents are not in accordance with the topic of this manuscript. Please clarify this.

Answer: Thank you for your suggestion. In addition to establishing a rapid RAA detection method, the analysis of phenotype and molecular characteristics for hvKp is also an important part in our manuscript. It can not only help us to verify that this RAA method can detect hvKp with different characteristics (serotype or STs), but also help us further understand the molecular characteristics of hvKp in China. In order to better describe our research, we revised the title to “Recombinase-Aided Amplification Assay for Rapid Detection of Hypervirulent *Klebsiella pneumoniae* (hvKp) and Characterization of the hvKp Pathotype”.

3. There is on hvKp/cKp standard strains were used to evaluate the specificity in the study. It is not enough to use only plasmid containing genes *peg-344* and *rmpA*, at least two standard strains of hvKp and cKp should also be examined.

Answer: Thanks for your suggestion. In this study, the *K. pneumoniae* strain ATCC BAA-2146 was used as reference strain for cKp. The *K. pneumoniae* strain LA.045 isolated from a liver abscess patient and proved to be hypervirulent using mouse lethality assay was used as reference strain for hvKp. As suggested, we have supplemented this information in the material method (Line285-288) and other corresponding parts of the manuscript.

4. It is not convincing to evaluate the RAA assay in clinical detection of hvKp only using clinical isolates. I suggest the authors to check the ability of the developed assay from clinical samples, such as, blood, saliva and nasopharyngeal

swabs.

Answer: Thanks for your suggestion. In the revised manuscript, we added an additional 50 *K. pneumoniae* negative clinical samples to evaluate our method. The samples include 10 fecal obtained from healthy individuals, 20 nasopharyngeal swabs, and 20 blood. For the *K. pneumoniae* positive clinical samples, we not only detected the isolates, we also tested DNA obtained from the original clinical samples. In the corresponding part of the manuscript, we have made modifications.

5. The authors should analyze the economics of this assay and explain in discussion, since the authors thought the cost of RAA assay is lower than real-time PCR or other nucleic acid-based detection methods in discussion.

Answer: Thanks for your suggestion. As there is no commercialized kit for hvKp detection at present, it is not rigorous to estimate the cost only by raw materials used in different methods. So, we deleted this sentence in discussion.

Minor comments:

1. Line 108: Delete the "peg-344 and rmpA"

Answer: Thanks for your suggestion. As suggested, we delete the "peg-344 and rmpA" (Line 111 in manuscript R1).

2. Line 114: Delete the "using the RAA assay developed"

Answer: As suggested, we delete the " using the RAA assay developed " (Line 117-118 in manuscript R1).

3. Line 114-116: Refine and rewrite this sentence.

Answer: As suggested, we refine and rewrite this sentence (Line 115-116 in manuscript R1).

4. Line 117: Change "The optimal reaction temperature range" to "The range of the reaction temperature".

Answer: As suggested, we change "The optimal reaction temperature range" to "The range of the reaction temperature" (Line 120 in manuscript R1).

5. Line 121: Change "Fluorescence" to "Fluorescent".

Answer: As suggested, we change "Fluorescence" to "Fluorescent" (Line 121 in

manuscript R1).

6. Line 123: Delete "evaluated"

Answer: As suggested, we delete "evaluated" (Line 124 in manuscript R1).

7. Line 126: Add "analysis" before the word "of"

Answer: As suggested, we add "analysis" before the word "of" (Line 126 in manuscript R1).

8. Line 130: The results from conventional PCR and real-time PCR could not be reviewed in Figure 3. Suggest the authors add these figures in Figure 3.

Answer: As suggested, we add the results from conventional PCR (Figure3B and 3E) and real-time PCR (Figure3C and 3F) in the revised Figure 3.

9. Line 131: Here, the unit of the concentration of recombinant plasmid is "copies/reaction", which is not in line with "copies/μl" in Line 135. Please the authors clarify.

Answer: As suggested, we revised "copies/reaction" to "copies/μl" in Line 128 (manuscript R1).

10. Line 141-143: Refine and rewrite this sentence.

Answer: As suggested, we refine and rewrite this sentence (Line 141-143 in manuscript R1).

11. Line 148: Delete the "assay" before the word "results".

Answer: As suggested, we delete the "assay" before the word "results" (Line 149 in manuscript R1).

12. Line 149: Delete "which".

Answer: As suggested, we delete "which" (Line 149-151 in manuscript R1).

13. Line 159-161: Refine and rewrite this sentence.

Answer: As suggested, we refine and rewrite this sentence (Line 159-162 in manuscript R1).

14. Line 198-199: Refine and rewrite this sentence.

Answer: As suggested, we refine and rewrite this sentence (Line 197-199 in manuscript R1).

15. Line 202-204: Refine and rewrite this sentence.

Answer: As suggested, we refine and rewrite this sentence (Line 201-204 in manuscript R1).

February 13, 2023

Prof. Jing Yuan
Capital Institute of Pediatrics
Department of Bacteriology
No. 2 yabao road, Chaoyang District
Beijing, Beijing 100020
China

Re: Spectrum03984-22R1 (Recombinase-Aided Amplification Assay for Rapid Detection of Hypervirulent *Klebsiella pneumoniae* (hvKp) and Characterization of the hvKp Pathotype)

Dear Prof. Jing Yuan:

Your manuscript has been accepted, and I am forwarding it to the ASM Journals Department for publication. You will be notified when your proofs are ready to be viewed.

Sincerely,

MARK PANDORI
Editor, Microbiology Spectrum
